# How Disclosure Types of Sustainability Performance Impact Consumers' Relationship Quality and Firm Reputation

**Warat Winit \***, **Erboon Ekasingh** and **Jomjai Sampet**

Faculty of Business Administration, Chiang Mai University, Chiang Mai 50200, Thailand
\* Correspondence: warat.winit@cmu.ac.th

**Abstract:** Given that firms attempt to gain competitive advantages from corporate sustainability schemes, we employed an experiment to examine different types of sustainable performance disclosure—output, outcome and impact—that best promote the quality of relationships with consumers, relationships with the firm around satisfaction and trust, and perceived reputation. Derived from a student sample of 254 respondents from a business school in Thailand, the findings indicate that, among different levels of sustainability performance disclosure, consumers are more likely to perceive the satisfaction, trust, and reputation of the firm as higher if the firm demonstrates the impact (rather than output or outcome) of sustainable performance. Results are consistent across observed product and service categories. Implications of findings and directions for future studies are also discussed.

**Keywords:** sustainability performance; disclosure types; output, outcome and impact; stakeholder focus; signaling theory; trust; corporate reputation

## 1. Introduction

Corporate sustainability (CS hereafter) has become one of the most relevant research issues nowadays. Motives that lead firms to conform to CS vary from mandatory to voluntary [1,2]. Specifically, while some firms comply with sustainability practices due to the pressure of government policies or regulations to prevent the future costs of social and environmental corporate irresponsibility, a number of firms are willing to co-operate with sustainability schemes voluntarily, as they expect to gain long-term competitive advantages such as increasing stock value [3], obtaining proactive leadership [4], enhancing trust [5], serving customer demands and expectations [4], and gaining reputation [6–8].

Stakeholder focus is one of the key philosophies of CS [9]. It suggests that the needs of relevant stakeholders should be firstly met and then the needs of shareholders would be later accomplished [10]. Firms attempt to meet the demands of their stakeholders due to underlying reasons that they need to avoid the possible pressures from stakeholders, as well as create a better society. Stakeholders act as a source of expectation about desirable and undesirable performance of the firm. They also evaluate how well a firm has met expectations and/or how firms' actions have impacted the groups in the environment [11].

To promote sustainability performance, evidences revealed that firms communicate these issues to stakeholders in different ways. Specifically, some firms highlight immediate results such as a list of projects launched or a list of organizations donated to. This type of performance is categorized as "output" of the performance. Moreover, some firms promote consequences of sustainable activities such as evidences of skill enhancement of local labor, amount of energy saved, and amount of carbon emission reduced. This is called the "outcome" of the performance. Additionally, some firms focus on the output of the project that serve as long-term benefits to broad range of stakeholders in society such as returning an eco-system to nature, achieving a zero-waste goal, becoming a leader in the green industry. This is called the "impact" of the performance [12,13].

With a limited body of existing knowledge, further research is needed to investigate a linkage between sustainability performance disclosure communication and perceptions among stakeholders. Peloza et al. [14] (p. 74) support this point by stating that "there is often a major gap between stakeholder perception and firm performance". That is, corporate communications nowadays overlook the fact that different stakeholders pay attention to different messages and communication methods. What and how stakeholders perceive may or may not align with performance highlighted by the firm. Moreover, when presenting CS performance, past studies and firms usually present information such as a checklist of practices, dialogue responding to stakeholders needs, amount of funds that a firm has donated and sponsorship of a broad range of stakeholders [15–18], third-party ranking [14], overall competitive performance and market share [18,19], or self-evaluated corporate performance such as financial performance [9], interchangeably, without concern about the mix of the message categories between output, outcome, and impact. As a consequence, a more comprehensive understanding of the relationship between different types of sustainability performance disclosure communication and stakeholder's attitudes is needed.

The present study, therefore, proposes that a firm has to start from having sustainability embedded in their vision to be sustainable. Then, it has to pay attention to delivering the right sustainability performance messages to the right stakeholders. A stakeholder-perceived sustainability performance message could lead to a stronger relationship and, consequently, enhanced corporate reputation. Corporate reputation becomes increasingly important as a form of CS performance outcome [1,2] which is associated with the capacity to deliver public benefits [3]. Therefore, corporate leaders need to thoroughly understand how stakeholders evaluate messages offered by a firm and how they react to the firm in return.

This recent study aims to focus on consumers as a unit of analysis. Past studies have extensively examined the relationship between perceived firm performance, relationship quality, brand equity, and behavioral intention among consumers [4–7]. Thus, it is interesting to further examine whether different types of sustainability performance disclosure (output, outcome, and impact) could lead to different degrees of perceived relationship quality, and brand reputation.

Participants used in this study were business students from a university in Thailand. Thailand is an appropriate context to examine the effect of sustainability disclosures on consumers as, among its top 100 companies by revenue, the sustainability reporting rate increased from 84% in 2020 to 97% in 2022 [8]. Moreover, Thailand is ranked among the top 10 countries worldwide in various aspects, including sustainability information, in annual financial reports, reporting against stock exchange guidelines, seeking assurance on sustainability reporting, and having sustainability representation at a leadership level [8]. Moreover, the Securities and Exchange Commission, Thailand has amended the regulations for Thai listed companies to prepare "One report" to improve disclosure efficiency under the Environmental, Social and Governance (ESG) principles [9]. These evidences highlight the influence of sustainability disclosure practice in Thailand. Therefore, using Thai business students to evaluate the impact of sustainability information disclosed in the reports of Thai listed companies in this study is appropriate.

The objectives of this study are threefold. First, it aims to demonstrate that consumers could differentiate sustainability performance information into three types, output, outcome and impact. Second, it will demonstrate that consumers who perceive different types of sustainable performance disclosure will exhibit their satisfaction and trust towards the brand differently. Third, it will demonstrate that consumers who perceive different types of sustainable performance disclosure will exhibit their perceived corporate reputation differently.

Findings in this study contribute to the CS study that types of disclosure could affect stakeholder relationships differently. Further, it provides guidance for marketing and brand

communication to develop effective communication strategy in order to gain a positive response from consumers.

## 2. Literature Review

### 2.1. CS Performance

CS is the micro-level of the sustainable development concept [10]. Focusing on the corporate dimension, CS strategies are increasingly deployed by many advanced companies all over the world [8,11]. However, there is no consensus on the definition of CS. For instance, Rogers et al. [12] have defined CS as the ability of a firm to balance financial, social, and environmental performance outcomes, considered as the triple bottom line (TBL). Kantabutra and Avery [13] defined CS as the ability of a firm to deliver strong financial performance, endure economic and social crises, and maintain a market leadership over time. In addition, Avery and Bergsteiner [14] have further suggested that a sustainable firm is a firm performance that has the ability to enhance customer satisfaction, demonstrates solid financial and operational performance, focuses on creating long-term shareholder and stakeholder values, and has excellent brand and reputation.

Among sustainable development schemes, corporate social responsibility (CSR) is another micro-level of sustainable development that focuses on the social dimension [10]. However, recent research strongly suggested that the role of CSR should be extended further to corporate responsibility, as the firm must shift its responsibility from just social and environmental issues to broader operational processes and business strategies [15]. A business needs to meet its sustainable profitability goal as well as participate in community matters wherever it is operating [16] by adopting four main CSR theories: (1) Instrumental theory, in which a firm has to attain long-term profits and competitive advantages; (2) Political theory, in which a firm has to utilize business power in a responsible way; (3) integrative theory, in which a firm has to integrate social demand into the business operation; (4) Ethical theory, in which a firm has to contribute ethically correct things to society [17].

### 2.2. Consumers and Firms' Stakeholder Focus

Recent literature relating to the stakeholder focus indicates that a firm has to fulfill benefits to its wide range of stakeholders prior to its own, in order to achieve sustainability performance outcomes [1,3,18,19]. To support this point, Economy of Communion literature also encourages the business world nowadays to have sharing, fellowship, fraternity, and reciprocity behaviors [20]. When the firm delivers benefits to stakeholders, stakeholders will establish satisfaction, trust, commitment, and identification to the firm in return. Then, the reciprocal relationship between the firm and its stakeholders is developed [21]. As a consequence, stakeholders will gain benefits and generate wealth back to the firm [5,18,21,22].

Business and society are inter-reliant [16]. CSR activities that promote CS benefits to stakeholders could consequently enhance the relationship between stakeholders and a firm [23–25], protect the reputation of a good enterprise [17,26,27], and subsequently lead to perceived positive reputation and brand equity of the firm [28–30] among stakeholders. In contrast, companies might suffer from reputation damage if they are unable to deliver the expected CS outcomes [26]. Thus, business should promote long-term benefits for stakeholders in society as a whole [16].

However, different groups of stakeholders have different needs/wants which create conflicts of interest in some situations [31]. For instance, while customers pay more attention to the quality and performance of the product, investors may focus mainly on the cost reduction and return on investment. Therefore, a firm needs to thoroughly understand stakeholders' needs and offer or allocate benefits that stakeholders perceive as valuable and fair to them. That is, the stakeholder-focus concept needs to shift the processes of stakeholder-oriented practices to the stakeholder's point of view instead of sticking with the perspective from the firm [32].

However, studies nowadays tend to overlook the role of consumers in stakeholders' focus on CS. For instance, "While the adoption of social causes by organizations has often been based on the assumption that consumers will reward this behavior, it is unlikely that consumers will blindly accept social initiatives as sincere, and so may or may not reward the firm with positive attitudes and purchases" [33]. This further proves that sustainability performance could lead to a consumer relationship with the company, and contributions to reputation should be thoroughly examined [5,34].

### 2.3. Output, Outcome, and Impact as Sustainability Performance Disclosure Types

Output, outcome, and impact have long been represented as the performance measurement or practice indicator of the firm [35–37]. *Output* could be described as the immediate effects or results set down in writing, a management plan, a permit, a law, etc., at the end of the decision-making process [35–37].

Outcome is defined as direct changes in human perceptions or consequent actions of the output, considered as mid-range or intermediate effects [35–37].

Impact is defined as long-term benefits to the firm and overall society or changes in the environment, accumulated from intended and unintended effects of the outcome [35–37]. It could be exhibited as the status of a leader in the industry [36].

Research in performance measurement generally has some unclear boundaries between output, outcome, and impact. Specifically, some firms demonstrate the mix between sustainable outputs and outcomes, e.g., number of trees planted, amount of donations, numbers of CS initiatives [38], etc. More importantly, impact measures seem to be misled, as a number of firms claim impact performance simply from self-reporting activities [39], rather than being recognized by the overall society.

This recent research argues that, if the firm aims to improve the well-being of the surrounding community, the output that the firm should aim for is to initiate community development plans, launch a local staff recruitment program, sponsor local community events, and fund educational scholarships for local members. Additionally, the increased number of local staff hired and income per head, positive feedbacks from community event participants, the increased number of graduated workers in the community should be considered as sustainable outcomes from the practice. Additionally, the impact of sustainable practices should be reported in terms of the following: being recognized as the top company among the community, achieving some awards relating to community relations and human resource management.

### 2.4. Relationship Quality

Bhattacharya et al. [21] (p. 263) have defined relationship quality as "overall assessment of the strength of a relationship, conceptualized as a composite or multidimensional construct capturing the different but related facets of a relationship." A reciprocal relationship exists between a firm and its stakeholders [18]. When the firm delivers benefits to stakeholders, stakeholders then develop a relationship with the firm in return to gain benefits and to generate wealth to the firm [22]. This reciprocal approach is supported by the Economy of Communion literature. It has stated that it is possible for the business world nowadays to have sharing, fellowship, fraternity, and reciprocity behaviors [20]. Relationship quality can be categorized into four levels which are satisfaction, trust, commitment and identification, depending on types of benefits that stakeholders receive from the firm [23–25]. The relationship quality develops over time from the lower level (satisfaction) to a higher level (identification). Since this study focuses on consumers' response to performance communication messages, it observes the development up to the initial levels of relationship quality. Thus, only satisfaction and trust are adopted in this study as consequences of perceived CS performance communication.

First, satisfaction is the overall evaluation regarding a firm/organization from stakeholder experience [21]. In general, stakeholders evaluate their satisfaction by comparing overall experience gained from a firm with resources that they have to contribute in order

to have a relationship with the firm [21]. Satisfaction is salient when perceived performance of a product/service meets the expectation of an individual [40].

Secondly, trust is defined as a perception of confidence in reliability and integrity among business partners or persons who come to interact with a firm [21]. Trust can be expressed as a form of stakeholder expectation that an organization will achieve what they promise, including perceived benevolence and not acting opportunistically towards stakeholders.

### 2.5. Perceived Corporate Reputation

Corporate reputation is a key factor that describes the success of a firm as viewed by stakeholders. Corporate reputation is defined as the perceptual representation of a firm's action in the past and in the future, which reflects the overall appeal of the firm to all key parties when comparing to other rivals [41]. Corporate reputation responds to the notion that sustainability performance outcomes and effectiveness of stakeholder-oriented practices of a firm should be evaluated by not only the firm but also its stakeholders [2]. Studies in the past have found that stakeholders' perceived social performance of a firm is one of the most important determinants of corporate reputation [42,43]. Avery and Bergsteiner [14] suggest that CS be earned through achieving excellent brand and reputation. Similarly, some studies have found that corporate reputation and brand equity are outcomes of CSR [28] and good reputation could lead to persistent profitability and sustaining superior performance [15]. Moreover, benefits that a firm receives from cause-related activities are in the form of stronger brand positioning, increased brand favorability, increased sales volume, enhanced customer loyalty, and building a strong relationship with alliances and social institutions [44].

### 2.6. Sustainability Performance Disclosure Types as Signal of Relationship Quality and Reputation Evaluation

Consumers' evaluation of output, outcome, and impact could be explained by signaling theory. Signaling theory addresses the recipients' evaluation of products, services, or brands based on relevant information deficit between parties through signals in order to reduce information asymmetry [45–47].

Organizational performances are associated with quality of signals. Firm management attempts to deliver positive sustainability signals such as competitive strategy, strong financial performance, and firm's stability status to relevant stakeholders (including consumers) so that high sustainable performance could signal high-quality information, resulting in reducing information asymmetry to recipients [48]. Reactions towards sustainable reporting signals could be categorized into three types (1) nonfinancial result, (2) investment decisions, and (3) reaction towards stock market [49]. As stakeholder theory indicates that different stakeholders may have different needs/wants, the methods of measuring outcomes for different stakeholder groups are accordingly varied. For instance, investment and reaction towards the stock market can be effectively used to measure sustainable outcomes among investors, shareholders and even the company, which is considered an internal stakeholder [50]. Additionally, nonfinancial outcomes such as satisfaction, trust, and corporate reputation are suitable outcome measures for consumers [49].

Among sustainability signals, a number of studies have investigated the relationships between various sustainability signals and consumers' evaluation. For instance, Atkinson and Rosenthal [51] have found that eco-label is a signal to affirm the credibility of environmental claims. Baumgartner et al. [52] have found that positive signaling of corporate reputation disclosure has positively affected organizational performance, corporate reputation, and stakeholders' intention. Bae et al. [48] found that signals from corporate governance elements, such as characteristics of shareholders, have positively affected sustainability disclosure. Friske et al. [49] have found that voluntary sustainability reporting is positively related to firm value. It promotes signals of transparency and accountability of the firm among publics. In contrast, environmental disclosure revealed a negative signal of profitability output [53].

Recently, CSR studies focusing on consumers found that CSR and sustainability schemes could act as signals to enhance evaluations of both product and corporate brands, satisfaction, trust, loyalty, brand admiration and brand equity [54]. In evaluating the clues of satisfaction, trust, and reputation of the firm, potential consumers lack complete information about the sustainability performance. To resolve information asymmetries, consumers tend to find information from signalers that clearly communicate benefits or values that they would gain from experiencing products or services that linked to signals of high quality, positive corporate reputation, and image [54,55]. However, it is still unclear regarding the underlying and unobservable qualities of the firm that the CS signals should demonstrate [56]. In addition, perceived long-term reputation needs time to be developed. Over time, CS signaling process may not be effective if consumers, considered as receivers, are not aware of what to look for or are not looking for the CS signals [54,55].

Studies in the past have found that third-party signals such as reputation lists, rating, and corporate brand rankings (such as such as the "100 Best Corporate Citizens," "America's Greenest Companies," and "World Most Admired Companies") are considered as effective signals as they contain following properties: (1) authentic signal, (2) costly to imitate, (3) consistent and clear (4) informational cues that aid in diagnosing the brand [49,54,57].

In regard to using awards or rankings as a social responsibility indicator, marketing literature, i.e., Rossi and Rivetti [58], found that third-party labels did not influence consumers' perceptions and their willingness to buy or pay. Baier, Göttsche, Hellmann, and Schiemann [59] have supported that reference explicitness must be presented along with depth of assurance for sustainability reporting to gain higher credibility. This is because high reference explicitness solely is interpreted as a misleading or false signal among consumers. However, accounting and economic literature found positive results for effects of disclosing CSR awards of rankings on financial performance.

Prior studies examined the relationship between CSR awards and corporate financial performance of listed companies in Thailand [60] and Taiwan [61]. They both found a significant and positive relationship between CSR awards and corporate financial performance. Awards or rankings given by a third party were also used as an indicator for strong environmental management. Klassen and McLaughlin [62] found a significant positive stock return for the 500 largest publicly traded US corporations with strong environmental management, measuring by environmental performance awards. The effect of eco-friendly certificates and awards on consumers' perceived value were also investigated. Lee et al. [63] found that such certificates and awards positively impact customers' perceived value within the hotel industry, which resulted in increased customers' satisfaction, retention, and intention to pay a green premium. However, Rossi and Rivetti [58] found no significant relationship between third-party sustainable labels and receiver's corporate evaluation per se, but found a significant relationship between a self-declared sustainable claim and the receiver's corporate evaluation.

Therefore, mixed results were found. Specifically, marketing scholars reported a negative interaction, whereas finance scholars reported a positive interaction [57], suggesting that further studies should be thoroughly examining each direct signal and third-parties signal separately.

## 3. Conceptual Model and Hypotheses

Based on the literature review, the following conceptual model is derived (see Figure 1). This present research posits that, while output, outcome and impact altogether are key sustainable performance information for managers, company's auditors and investors, impact is the key signal for consumers that enhances satisfaction, trust, and corporate reputation.

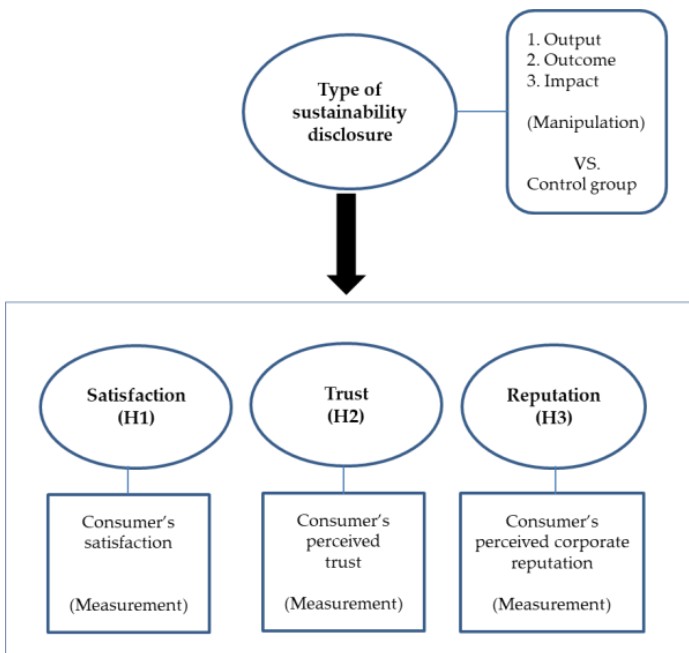

**Figure 1.** Conceptual model.

More precisely, we posit that, when evaluating the CS, consumers tend to acquire relevant information from trusted sources that clearly communicate direct benefits to them. Output and outcome signals are based on internal performance evaluation of the firm and are less associated with value exchanged with consumers. Impact, in contrast, is the trusted sources from third-party agents that provide comparative sustainability signals among competitors, in the form of ranking, and ratings. Thus, consumers could compare and associate the benefits of impact directly with their observed product and brand. Perception of positive impact will consequently influence a stronger relationship with the firm in the forms of brand satisfaction, brand trust, and perceived corporate reputations. Three hypotheses are also proposed as follows:

**H1.** *Among different levels of sustainability information disclosure, consumers are more likely to perceive the satisfaction of the firm as higher if the firm demonstrates impact (rather than output or outcome) of the sustainable performance report.*

**H2.** *Among different levels of sustainability information disclosure, consumers are more likely to perceive trust of the firm as higher if the firm demonstrates impact of the sustainable performance report.*

**H3.** *Among different levels of sustainability information disclosure, consumers are more likely to perceive the reputation of the firm as higher if the firm demonstrates impact of the sustainable performance report.*

## 4. Research Methodology

Quasi-experimental research was applied with university students, based on a $3 \times 1 + 1$ (control group) between-subjects design. The method used in this study was consistent with Baier et al. [59]. The independent variables (IVs) were comprised of three information types of sustainable performance reporting, which were output, outcome, and impact, across four product and service brands. Dependent variables (DVs) were comprised of satisfaction, trust, and corporate reputation.

This study, therefore, is divided into two phases: the preliminary study and the main study. The objective of the preliminary study is threefold. First, it aims to develop information types of sustainable performance stimuli. Second, it aims to select appropriate brands of product and service that suit the sample and context of study. Third, it aims to develop and pretest dependent measures that are suitable to a Thai context. The main

study was adapted from [35]. It aims to investigate the roles of different information types of sustainable performance reported (output, outcome, and impact) on consumers' satisfaction, trust, and perceived corporate reputation. Details of both phases will be discussed as follows.

*4.1. Preliminary Study*

4.1.1. Brand Selection

Four companies from different industries were chosen to examine the variations between high–low involvements of products and services. Specifically, the SCG construction materials company represents a high involvement–tangible product, SCB Bank represents a high involvement–intangible service, the Thai Union processed seafoods company represents a low involvement–tangible product, and True communication company represents a low involvement–intangible service. These four companies were selected because the samples equally have familiarity and access to use. These companies are also listed in the Thai Stock Market, with international recognition, and all of them provide all three sustainable information types in their annual or sustainable reports; thus, reducing the possibility of bias.

4.1.2. Sustainable Performance Stimuli

Experimental manipulation procedures were consistent with Baier et al. [59], Thakor and Lavack [64] and Winit et al. [65]. Based on a literature review [35–37], three information types of sustainable performance report were developed as experimental stimuli. A secondary information search was conducted through each company's public report such as the annual report, sustainability report, or public relations materials, deriving three types of information cues. Corporate-fact information was manipulated to achieve sustainable performance output, outcome, and impact information types. Sustainable performance information was provided in the form of a brand logo, corporate background (File S1 in supplementary), as well as a corporate-facts table consisting of sustainable performance evidences that relate to four key stakeholders—customers, employees, environment, and community.

More importantly, each corporate-fact table is manipulated to represent each type of sustainable information condition as follows (File S2). For the output condition, details in the table provided only what sustainable activities the firm has carried out for each group of stakeholders. Outcome condition provides the numerical/statistical result of what the firm has carried out in terms of sustainable activities. Impact condition provides the world and the regional sustainability-related rankings/ratings that the firm achieved. Lastly, the control condition provides only brand logo and corporate background with no corporate-fact presence. In addition, all other relevant product variables were kept constant across product/service categories as well as their financial performance.

Four versions of online questionnaire were applied, consisting of output, outcome, impact, and control versions. Each questionnaire version consisted of four brand scenarios in a single information type of sustainable performance report condition. Eight versions were developed, according to four information types, with two brand sequences for each to access ordering effects (Sequence 1, arranged as SCG, Thai Union, SCB, and True; and sequence 2, arranged as True, SCB, Thai Union, and SCG).

After exposure to corporate-fact information, each respondent was asked to complete a manipulation check by indicating their perception of sustainable performance information type (File S3). It was expected that respondents who were under the output condition would agree with the statement that "Information of firms above indicating the policy, plan, and activities of the organization without providing the consequence results" while respondents who were under the outcome condition would agree with the statement "Information of firms above indicating the policy, plan, and activities of the organization with statistics showing consequence results". In addition, respondents who were under the impact condition were expected to agree with the statement that "Information of firms

above indicating the policy, plan, and activities of the organization with awards, ranking, or certification showing long-term results".

### 4.1.3. Dependent Measures

Satisfaction towards brand consisted of three items using seven-point semantic differential scales from Winit and Kantabutra [34], He and Li [66]. Perceived brand trust consisted of five items using seven-point semantic differential scales from Winit and Kantabutra [34], Morgan and Hunt [67]. Perceived brand reputation measures from Winit and Kantabutra, and Hsu [28,34] were adopted, comprising five items (File S4).

Scales were translated into Thai by a translator who was fluent in Thai and English, and then back-translated by a bilingual expert to verify the correctness of the conceptual translation and to confirm that the scale items properly fit the Thai context. Minor corrections were made to enhance consistency with the original English version. Thai measures were then pretested to verify interpretation clarity and scale reliability.

### 4.1.4. Preliminary Study Results

Among 46 students, 36 correctly perceived manipulation conditions accounting for 78.26%. Respondents correctly perceived manipulation in each condition over 70%, suggesting that the manipulation was successful and could apply in the main study. Table 1 column I demonstrates the results of the manipulation check.

**Table 1.** Manipulation check.

| Conditions | Manipulations | I: Preliminary Study | | II: Main Study | |
|---|---|---|---|---|---|
| | | Frequency | Percent | Frequency | Percent |
| Output | Fail | 2 | 13.3 | 10 | 13.5 |
| | Correct | 13 | 86.7 | 64 | 86.5 |
| | Total | 15 | 100 | 74 | 100 |
| Outcome | Fail | 4 | 25 | 16 | 19.5 |
| | Correct | 12 | 75 | 66 | 80.5 |
| | Total | 16 | 100 | 82 | 100 |
| Impact | Fail | 4 | 26.7 | 25 | 27.78 |
| | Correct | 11 | 73.3 | 65 | 72.22 |
| | Total | 15 | 100 | 90 | 100 |
| Control | | | | 59 | |

### 4.2. Main Study

In the main study, an online quasi-experiment was conducted to examine how consumers' satisfaction and trust vary across output, outcome, and impact conditions. Thus, these IVs were manipulated as $3 \times 1$ between subject design. Details of the main study are described below.

### 4.2.1. Samples

The samples employed for the main experiment were 310 students from a university in Chiang Mai, the second largest city of Thailand. Homogeneity characteristics of students support more precise predictions, resulting in a stronger test of a given scope of study or theory. The recruited business students were volunteers from various majors and stages of their program. Their ages were between 18 and 25 years old, representing young, urban and educated consumers which fit the general characteristics of opinion leaders. These students with similar age-range had similar brand familiarity, brand awareness, or chances to purchase or use services of all four companies; therefore, they were appropriate for achieving the study's objectives. Baier et al. [59] have suggested that a student sample is justified, in their study examining the relationship between sustainability reporting and consumer reactions. This is because a variable assessing the report, such as credibility, does not require participants to draw complex connections and students in business are a proxy

for reasonably informed non-professional investors; thus, equivalent to consumers. More importantly, Verlegh and Steenkamp [68] reported no differences in the magnitude between studies employing student samples and those utilizing consumer samples.

4.2.2. Data Collection and Data Analysis Methods

Data collection occurred in the classroom setting. With prior permission from the lecturer for the class period, participants were recruited for volunteers who were willing to fill out the online survey questionnaire through their mobile or computer devices, which took 15 to 20 min to complete. The researcher and the lecturer informed participants that this was to be purely on a voluntary basis and was not mandatory and would have no impact whatsoever on their course grade. The participants were also allowed to withdraw from the survey at any time during and after the administration of the questionnaire. Additionally, participants who completed the questionnaire received a 200 Baht (7 USD) gift card as a token of appreciation for their time.

All documents, consisting of (1) a letter of consent, (2) a letter of instruction, and (3) the questionnaire, were then distributed to the respondents via QR codes. At least one researcher (or research assistant) attended each session to facilitate and monitor the survey. He/she was trained and briefed thoroughly so that he/she could answer all questions raised by participants (if necessary). All participants received the same instructions for each treatment. They were also asked to read and provide informed consent before completing the questionnaire. They were informed that, as consumers, they would be asked to provide opinions on the performance of four large companies. They were then instructed to begin reading the case materials and complete the questionnaire. The case materials provided a brief company background information along with the company's key stakeholders and CSR performance. Once the participant completed the questionnaire, they were asked to submit the questionnaire, complete the debriefing, and leave the online session. Data collected were automatically stored in the researcher's Google drive (in a password-encrypted computer).

Amos was employed for tests of dimensionality of measures. Analysis of variance (ANOVA) was employed for examining the differences of satisfaction, trust, and corporate reputation across four manipulation conditions. Specifically, H1, H2, and H3 were supported if the mean of satisfaction, trust, and perceived corporate reputation scores in the impact condition were statistically higher than in the control condition, across different product and service categories.

## 5. Results

*5.1. Manipulation Check*

Among 305 students, 254 (83.28%) correctly perceived manipulation conditions. Respondents correctly perceived manipulation in each condition over 70%, suggesting that the manipulation was successful and could be applied in the main study. Table 1 column II describes the results of the manipulation check.

Three dependent variables (satisfaction, trust, and reputation) revealed a good fit across four firms. Cronbach's alpha of all constructs was higher than 0.80, suggesting that each variable is appropriate for further analysis [69,70]. More details about Cronbach's alpha and fit indices are provided in Table S1, File S5.

Table 2 reports satisfaction, trust, and perceived brand reputation means for the three groups regarding information types of sustainable performance (output, outcome and impact) and the control group. As expected, for all companies, respondents in an impact-information-type group report more satisfaction, trust and perceived brand reputation than ones in the other three groups (output, outcome and control).

**Table 2.** Mean satisfaction, trust, and reputation; descriptive.

| Firm | Conditions | n | Constructs | | | | | |
|---|---|---|---|---|---|---|---|---|
| | | | I: Satisfaction | | II: Trust | | III: Reputation | |
| | | | Mean | SD | Mean | SD | Mean | SD |
| SCG | Output | 64 | 4.9271 | 1.05153 | 4.8625 | 1.12511 | 5.8219 | 0.97337 |
| | Outcome | 66 | 5.0101 | 1.04099 | 5.0273 | 1.11928 | 5.8242 | 0.84089 |
| | Impact | 65 | 5.3077 | 0.92854 | 5.4123 | 0.99930 | 6.1938 | 0.73312 |
| | Control | 59 | 4.6893 | 0.87945 | 4.8814 | 0.90905 | 5.7864 | 0.83366 |
| | Total | 254 | 4.9908 | 0.99864 | 5.0504 | 1.06232 | 5.9094 | 0.86082 |
| TU | Output | 64 | 4.2396 | 0.75496 | 4.1656 | 0.88916 | 4.7031 | 0.91547 |
| | Outcome | 66 | 4.4848 | 0.85171 | 4.5394 | 0.96968 | 4.9848 | 0.93566 |
| | Impact | 65 | 4.5385 | 0.90847 | 4.7538 | 0.99735 | 5.2123 | 0.99679 |
| | Control | 59 | 4.0508 | 0.76015 | 4.0237 | 0.77378 | 4.5559 | 0.85183 |
| | Total | 254 | 4.3360 | 0.84087 | 4.3803 | 0.95284 | 4.8724 | 0.95650 |
| SCB | Output | 64 | 4.8281 | 1.24296 | 4.8219 | 1.36399 | 5.3219 | 1.13787 |
| | Outcome | 66 | 4.9545 | 1.00066 | 4.9545 | 1.21569 | 5.5182 | 1.02100 |
| | Impact | 65 | 5.2769 | 1.09856 | 5.3354 | 1.01311 | 5.8062 | 0.84777 |
| | Control | 59 | 4.5706 | 1.31605 | 4.5729 | 1.39727 | 5.1898 | 1.17175 |
| | Total | 254 | 4.9160 | 1.18657 | 4.9299 | 1.27486 | 5.4661 | 1.06784 |
| TRUE | Output | 64 | 4.1042 | 1.31049 | 3.9125 | 1.33660 | 4.9271 | 1.05153 |
| | Outcome | 66 | 4.3283 | 1.18321 | 4.1697 | 1.27810 | 5.0101 | 1.04099 |
| | Impact | 65 | 4.5282 | 0.92963 | 4.5231 | 1.15850 | 5.3077 | 0.92854 |
| | Control | 59 | 3.9661 | 1.10847 | 3.9119 | 1.18861 | 4.6893 | 0.87945 |
| | Total | 254 | 4.2388 | 1.15462 | 4.1354 | 1.26138 | 4.9908 | 0.99864 |

*5.2. Results*

The mean difference test for all dependent measures (satisfaction, trust, and perceived brand reputation) between the four groups was tested using ANOVA. The results for all companies are presented in Table 3, in which column I is for satisfaction, column II is for trust, and column III is for perceived brand reputation. Results for all companies are statistically significant for all dependent measures. These suggest that, for all companies, respondents' satisfaction, trust, and perceived brand reputation are contingent upon information types of sustainable performance.

**Table 3.** ANOVA results for mean difference test.

| | Column I: Satisfaction | | | | Column II: Trust | | | | Column III: Brand Reputation | | | |
|---|---|---|---|---|---|---|---|---|---|---|---|---|
| | Sum of Squares | df | Mean Square | F (Sig.) | Sum of Squares | df | Mean Square | F (Sig.) | Sum of Squares | df | Mean Square | F (Sig.) |
| SCG | | | | | | | | | | | | |
| Between groups | 12.176 | 3 | 4.059 | 4.225 | 12.494 | 3 | 4.165 | 3.814 | 7.120 | 3 | 2.373 | 3.290 |
| Within groups | 240.136 | 250 | 0.961 | (0.006) | 273.021 | 250 | 1.092 | (0.011) | 180.357 | 250 | 0.721 | (0.021) |
| Total | 252.132 | 253 | | | 285.515 | 253 | | | 187.477 | 253 | | |
| TU | | | | | | | | | | | | |
| Between groups | 9.519 | 3 | 3.173 | 4.225 | 21.191 | 3 | 7.064 | 8.469 | 16.087 | 3 | 5.362 | 6.224 |
| Within groups | 169.368 | 250 | .667 | (0.006) | 208.510 | 250 | 0.834 | (0.000) | 215.380 | 250 | 0.862 | (0.000) |
| Total | 178.887 | 253 | | | 229.702 | 253 | | | 231.467 | 253 | | |
| SCB | | | | | | | | | | | | |
| Between groups | 16.097 | 3 | 5.366 | 3.944 | 18.994 | 3 | 6.331 | 4.036 | 13.530 | 3 | 4.510 | 4.101 |
| Within groups | 340.111 | 250 | 1.360 | (0.009) | 392.198 | 250 | 1.569 | (0.008) | 274.959 | 250 | 1.100 | (0.007) |
| Total | 356.208 | 253 | | | 411.193 | 253 | | | 288.489 | 253 | | |
| TRUE | | | | | | | | | | | | |
| Between groups | 11.520 | 3 | 3.840 | 2.947 | 15.975 | 3 | 5.325 | 3.444 | 15.850 | 3 | 5.283 | 4.752 |
| Within groups | 325.768 | 250 | 1.303 | (0.033) | 386.566 | 250 | 1.546 | (0.017) | 277.945 | 250 | 1.112 | (0.003) |
| Total | 337.288 | 253 | | | 402.541 | 253 | | | 293.795 | 253 | | |

Table 4, columns I, II and III, present the results of a post-hoc Tukey HSD test for satisfaction, trust, and perceived brand reputation, respectively. H1 states that, among

different levels of sustainability information disclosure, consumers are more likely to perceive the satisfaction of the firm as higher if the firm demonstrates impact (rather than output or outcome) of the sustainable performance report. Table 4, column I and Figure 2 exhibit the results of a mean difference test for satisfaction between the four groups. The results suggest that, for all companies, the satisfaction mean of the impact-information-type group is statistically higher than that of the control group and this result is consistent across companies (for SCG, sig. = 0.003; for TU, sig. = 0.006; for SCB, sig. = 0.005; for TRUE, sig. = 0.033).

**Table 4.** ANOVA showing the mean difference in satisfaction, trust, and perceived brand reputation between the four groups of information type.

| Conditions | | I: Satisfaction | | | II: Trust | | | III: Brand Reputation | | |
|---|---|---|---|---|---|---|---|---|---|---|
| | | Mean Difference | SE | Sig. | Mean Difference | SE | Sig. | Mean Difference | SE | Sig. |
| **SCG** | | | | | | | | | | |
| Output | Outcome | −0.08302 | 0.17194 | 0.963 | −0.16477 | 0.18333 | 0.805 | −0.00237 | 0.14901 | 1.000 |
| | Impact | −0.38061 | 0.17259 | 0.125 | −0.54981 | 0.18402 | 0.015 | −0.37197 | 0.14957 | 0.064 |
| | Control | 0.23782 | 0.17689 | 0.536 | −0.01886 | 0.18861 | 1.000 | 0.03543 | 0.15330 | 0.996 |
| Outcome | Output | 0.08302 | 0.17194 | 0.963 | 0.16477 | 0.18333 | 0.805 | 0.00237 | 0.14901 | 1.000 |
| | Impact | −0.29759 | 0.17126 | 0.306 | −0.38503 | 0.18261 | 0.153 | −0.36960 | 0.14842 | 0.064 |
| | Control | 0.32084 | 0.17560 | 0.263 | 0.14592 | 0.18723 | 0.864 | 0.03780 | 0.15218 | 0.995 |
| Impact | Output | 0.38061 | 0.17259 | 0.125 | 0.54981 | 0.18402 | 0.016 | 0.37197 | 0.14957 | 0.064 |
| | Outcome | 0.29759 | 0.17126 | 0.306 | 0.38503 | 0.18261 | 0.153 | 0.36960 | 0.14842 | 0.064 |
| | Control | 0.61843 * | 0.17623 | 0.003 | 0.53095 * | 0.18791 | 0.026 | 0.40741 * | 0.15273 | 0.040 |
| Control | Output | −0.23782 | 0.17689 | 0.536 | 0.01886 | 0.18861 | 1.000 | −0.03543 | 0.15330 | 0.996 |
| | Outcome | −0.32084 | 0.17560 | 0.263 | −0.14592 | 0.18723 | 0.864 | 0.03780 | 0.15218 | 0.995 |
| | Impact | −0.61843 * | 0.17623 | 0.003 | −0.53095 * | 0.18791 | 0.026 | −0.40741 * | 0.15273 | 0.040 |
| **TU** | | | | | | | | | | |
| Output | Outcome | −0.24527 | 0.14440 | 0.327 | −0.37377 | 0.16022 | 0.093 | −0.28172 | 0.16283 | 0.310 |
| | Impact | −0.29888 | 0.14494 | 0.169 | −0.58822 * | 0.16082 | 0.002 | −0.50918 * | 0.16345 | 0.011 |
| | Control | 0.18874 | 0.14855 | 0.583 | 0.14190 | 0.16483 | 0.825 | 0.14719 | 0.16752 | 0.816 |
| Outcome | Output | 0.24527 | 0.14440 | 0.327 | 0.37377 | 0.16022 | 0.093 | 0.28172 | 0.16283 | 0.310 |
| | Impact | −0.05361 | 0.14383 | 0.982 | −0.21445 | 0.15959 | 0.536 | −0.22746 | 0.16220 | 0.499 |
| | Control | 0.43400 * | 0.14747 | 0.019 | 0.51567 * | 0.16363 | 0.010 | 0.42892 | 0.16630 | 0.051 |
| Impact | Output | 0.29888 | 0.14494 | 0.169 | 0.58822 * | 0.16082 | 0.002 | 0.50918 * | 0.16345 | 0.011 |
| | Outcome | 0.05361 | 0.14383 | 0.982 | 0.21445 | 0.15959 | 0.536 | 0.22746 | 0.16220 | 0.499 |
| | Control | 0.48761 * | 0.14800 | 0.006 | 0.73012 * | 0.16422 | 0.000 | 0.65638 * | 0.16690 | 0.001 |
| Control | Output | −0.18874 | 0.14855 | 0.583 | −0.14190 | 0.16483 | 0.825 | −0.14719 | 0.16752 | 0.816 |
| | Outcome | −0.43400 * | 0.14747 | 0.019 | −0.51567 * | 0.16363 | 0.010 | −0.42892 | 0.16630 | 0.051 |
| | Impact | −0.48761 * | 0.14800 | 0.006 | −0.73012 * | 0.16422 | 0.000 | −0.65638 * | 0.16690 | 0.001 |
| **SCB** | | | | | | | | | | |
| Output | Outcome | −0.12642 | 0.20462 | 0.926 | −0.13267 | 0.21973 | 0.931 | −0.19631 | 0.18398 | 0.710 |
| | Impact | −0.44880 | 0.20539 | 0.130 | −0.51351 | 0.22056 | 0.094 | −0.48428 * | 0.18468 | 0.046 |
| | Control | 0.25750 | 0.21051 | 0.613 | 0.24899 | 0.22606 | 0.689 | 0.13204 | 0.18928 | 0.898 |
| Outcome | Output | 0.12642 | 0.20462 | 0.926 | 0.13267 | 0.21973 | 0.931 | 0.19631 | 0.18398 | 0.710 |
| | Impact | −0.32238 | 0.20382 | 0.391 | −0.38084 | 0.21887 | 0.305 | −0.28797 | 0.18326 | 0.397 |
| | Control | 0.38392 | 0.20898 | 0.258 | 0.38166 | 0.22441 | 0.325 | 0.32835 | 0.18790 | 0.301 |
| Impact | Output | 0.44880 | 0.20539 | 0.130 | 0.51351 | 0.22056 | 0.094 | 0.48428 * | 0.18468 | 0.046 |
| | Outcome | 0.32238 | 0.20382 | 0.391 | 0.38084 | 0.21887 | 0.305 | 0.28797 | 0.18326 | 0.397 |
| | Control | 0.70630 * | 0.20973 | 0.005 | 0.76250 * | 0.22522 | 0.005 | 0.61632 * | 0.18858 | 0.007 |
| Control | Output | −0.25750 | 0.21051 | 0.613 | −0.24899 | 0.22606 | 0.689 | −0.13204 | 0.18928 | 0.898 |
| | Outcome | −0.38392 | 0.20898 | 0.258 | −0.38166 | 0.22441 | 0.325 | −0.32835 | 0.18790 | 0.301 |
| | Impact | −0.70630 * | 0.20973 | 0.005 | −0.76250 * | 0.22522 | 0.005 | −0.61632 * | 0.18858 | 0.007 |
| **TRUE** | | | | | | | | | | |
| Output | Outcome | −0.22412 | 0.20026 | 0.678 | −0.25720 | 0.21815 | 0.240 | −0.39441 | 0.18498 | 0.146 |
| | Impact | −0.42404 | 0.20102 | 0.153 | −0.61058 * | 0.21897 | 0.006 | −0.57101 * | 0.18568 | 0.012 |
| | Control | 0.13806 | 0.20603 | 0.908 | 0.00064 | 0.22443 | 0.998 | 0.00302 | 0.19030 | 1.000 |
| Outcome | Output | 0.22412 | 0.20026 | 0.678 | 0.25720 | 0.21815 | 0.240 | 0.39441 | 0.18498 | 0.146 |
| | Impact | −0.19992 | 0.19948 | 0.748 | −0.35338 | 0.21729 | 0.105 | −0.17660 | 0.18425 | 0.773 |
| | Control | 0.36218 | 0.20452 | 0.290 | 0.25783 * | 0.22279 | 0.248 | 0.39743 | 0.18891 | 0.155 |
| Impact | Output | 0.42404 | 0.20102 | 0.153 | 0.61058 * | 0.21897 | 0.006 | 0.57101 * | 0.18568 | 0.012 |
| | Outcome | 0.19992 | 0.19948 | 0.748 | 0.35338 | 0.21729 | 0.105 | 0.17660 | 0.18425 | 0.773 |
| | Control | 0.56210 * | 0.20526 | 0.033 | 0.61121 * | 0.22360 | 0.007 | 0.57403 * | 0.18960 | 0.014 |
| Control | Output | −0.13806 | 0.20603 | 0.908 | −0.00064 | 0.22443 | 0.998 | −0.00302 | 0.19030 | 1.000 |
| | Outcome | −0.36218 | 0.20452 | 0.290 | −0.25783 | 0.22279 | 0.248 | −0.39743 | 0.18891 | 0.155 |
| | Impact | −0.56210 * | 0.20526 | 0.033 | −0.61121 * | 0.22360 | 0.007 | −0.57403 * | 0.18960 | 0.014 |

* Significant difference between variables.

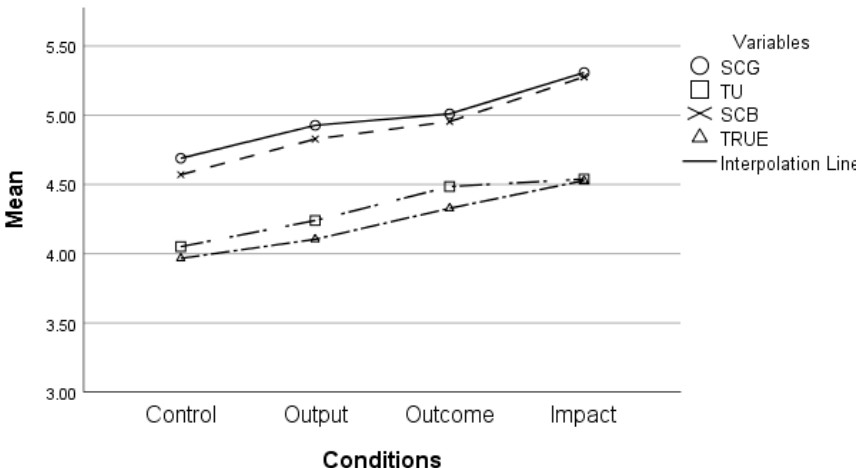

**Figure 2.** Mean of satisfaction variable across four conditions and product/service categories.

In contrast, mixed results were found among the output, outcome, and control groups. There is no difference in the satisfaction means among the output, outcome, and control groups, among SCG, SCB, and True. Although the satisfaction mean of the outcome group of TU is significantly higher than that of the control group (sig. = 0.019), the satisfaction mean of the impact group of TU is still the highest. Therefore, H1 is supported. These results suggest that, overall, the output and outcome information types do not affect respondents' satisfaction differently from only corporate-fact information (control). Respondents' satisfaction is significantly higher for the impact-information-type.

H2 states that, among different levels of sustainability information disclosure, consumers are more likely to perceive trust of the firm as higher if the firm demonstrates the impact of the sustainable performance report. Based on the results in Table 4, column II and Figure 3, the trust mean of the impact-information-type group is found to be statistically higher than that of the control group for all companies (for SCG, sig. = 0.026; for TU, sig. = 0.000; for SCB, sig. = 0.005; for TRUE, sig. = 0.034). Moreover, the trust mean of the impact is also statistically higher than that of the output group. These results are consistent with that of satisfaction. Similar to the mix results regarding satisfaction in column I, there is no difference in the trust means among output, outcome, and control groups for SCG, SCB, and True. Although the trust mean of the outcome group of TU is significantly higher than the control group (sig. = 0.01), the trust mean of the impact group of TU is still the highest. Therefore, H2 is supported.

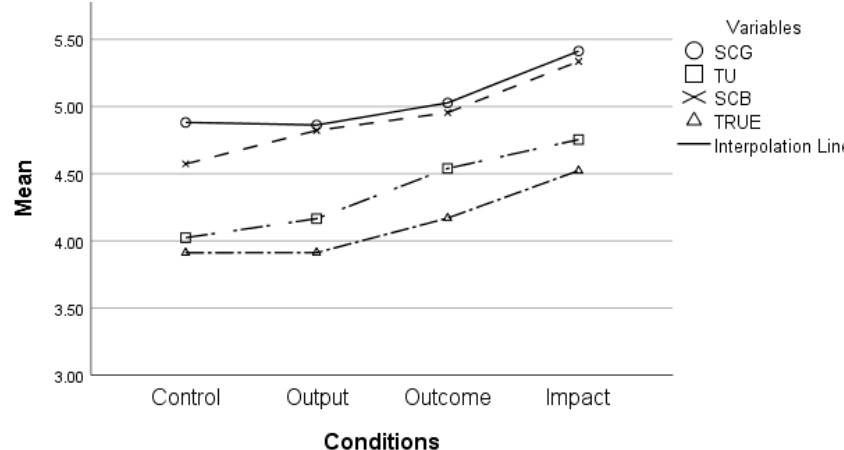

**Figure 3.** Mean of trust variable across four conditions and product/service categories.

H3 states that, among different levels of sustainability information disclosure, consumers are more likely to perceive the reputation of the firm as higher if the firm demon-

strates impact of the sustainable performance report. The results of the mean difference test for perceived brand reputation are shown in Table 4, column III and Figure 4. Consistent with the results in column I and column II regarding satisfaction and trust, respectively, the reputation mean of the impact information type group is significantly higher than that of the control group for all companies (for SCG, sig. = 0.008; for TU, sig. = 0.000; for SCB, sig. = 0.001; for TRUE, sig. = 0.003). As expected, the results concerning the mean difference among the output, outcome and control groups are mixed. Most of the means among output, outcome and control groups for all companies are not statistically different. The reputation mean of the output group is found to be significantly lower than that of the impact-information-type group for TU (sig. = 0.011), SCB (sig. = 0.046) and TRUE (sig. = 0.012). Therefore, H3 is supported.

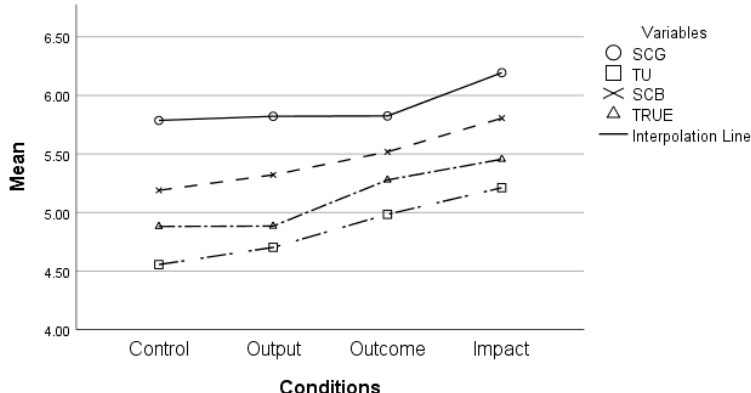

**Figure 4.** Mean of perceived reputation variable across four conditions and product/service categories.

## 6. Discussions

Based on sustainability performance literature in the past [35,37], this experimental study clearly points out that sustainability performance disclosure can be categorized into three types, namely output, outcome, and impact. More importantly, the result in this study successfully demonstrates that when students, considered as representative of consumers, are exposed to different types of sustainability performance disclosure, they indicate satisfaction, trust, and perceived corporate reputation differently, due to their different needs [31,33].

From consumers' point of view, impact is the most effective type of disclosure as it leads to the highest satisfaction, trust, and perceived corporate reputation, supporting H1, H2, and H3. In contrast, consumers indicate indifference when exposed to other forms of sustainability performance such as output and outcome, comparing to none of the information exposed (controlled group). Results revealed consistency across product and service industries.

Findings of this study could be explained by the signaling theory [45,47]. While findings in Baier et al. [59] are in contrast to their hypotheses, results in the current study could be explained by Baier et al.'s [59] framework. It extends the signaling theory research in CS disclosure [48,52] by further suggesting that consumers perceive impact information such as the status of being a leader in the industry, described by outstanding score/rankings given by a third-party organization, as the most effective signal for evaluating high performance of the firm, under limited time, knowledge, and resources provided. Third-party rankings and ratings of impact are considered as assured information because this information is judged by experienced or professional organizations. They contain an authentic signal, costly to imitate, consistent and clear, and provide useful informational evaluating cues as suggested by [49,54,57]. These impact properties manipulated in the study also reflect incremental time and effort of the firm to develop and maintain depth of assurance from third-party organizations and reference explicitness; thus, supporting suggestions made by Baier et al. [59]. In doing so, a firm will gain stronger corporate reputation which is positively related to firm value [49]. The result also provides insight into the

study of Friske [49] that, among consumers, nonfinancial results such as satisfaction, trust, and corporate reputation are effective criteria for evaluating sustainable performance of the firm.

## 7. Conclusions

In conclusion, this study sheds light on the relationship between different types of sustainability performance communication and consumers' satisfaction, trust, and perceived corporate reputations. In term of theoretical implications, this study fills the signaling research gap suggested by [61]. Results from the manipulation procedure revealed important unobserved CS signals, namely output, outcome, and impact. Findings in this study comprehensively bridge the gap between consumer perception and firm performance raised by [14], as they suggest that "impact" is the most effective way to communicate a firm's performance to consumers.

In terms of practical implications, the results also recommend a change in the communication norm of the past, that the firms mix the message categories between output, outcome, and impact when targeting consumers. Specifically, instead of spending their budget on a broad range of sustainable performance messages, a firm should omit unnecessary message types and focus on "impact", as sustainable performance content that consumers perceive value in and that aligns with performance highlighted by the firm. In doing this, the firm could reduce the overwhelming nature of signal information, use their communication budget more effectively, and gain the value of an effective sustainable performance signal.

While Wichianrak et al. [53] found a negative relationship between the signals of environmental disclosure and firm profitability, consumers in this study evaluated sustainability performance as an inclusive signal. Future research should thoroughly examine consumers' evaluations by separating signals into environmental, societal, and profitability aspects to gain insight into which dimensions of the core sustainability concept are the most important from the consumer's point of view.

**Supplementary Materials:** The following supporting information can be downloaded at: https://www.mdpi.com/article/10.3390/su15010803/s1.

**Author Contributions:** Conceptualization, W.W. and E.E.; Methodology, W.W. and E.E.; Formal analysis, W.W. and J.S.; Resources, E.E.; Data curation, W.W., E.E. and J.S.; Writing—original draft, W.W.; Writing—review & editing, E.E. and J.S.; Project administration, W.W. All authors have read and agreed to the published version of the manuscript.

**Funding:** This research was funded by Faculty of Business Administration, Chiang Mai University And The APC was funded by Center for Research on Sustainable Leadership, College of Management, Mahidol University.

**Institutional Review Board Statement:** The study was conducted in accordance with the Declaration of Helsinki, and approved by the Ethics Committee of CHIANG MAI UNIVERSITY (protocol code 64/032 and date of approval is 19 April 2021).

**Informed Consent Statement:** Informed consent was obtained from all subjects involved in the study.

**Data Availability Statement:** Data is unavailable due to ethical restriction.

**Conflicts of Interest:** The authors declare no conflict of interest.

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
