# Peer review of "How Disclosure Types of Sustainability Performance Impact Consumers’ Relationship Quality and Firm Reputation"

_sustainability, doi:10.3390/su15010803_

Round 1

Reviewer 1 Report

The title and abstract are well written

My recommendations are:

1. the abbreviations should not be followed by the word hereafter as in line 25

2. line 60, you already explained the abbreviation, so use only the abbreviation. Delete corporate sustainability in words there. Also line 96. Check the entire paper please

I liked that you presented both the introduction and literature review. The literature review is really extensive. I think it could have been comprised but I do not have a problem with this. 

Also, I liked the detailed explanations for the method used. Also the preliminary part. 

3.  At 4.2.2. Capitalize word Data if you used capital letter for the other words

4. Lines 434-435, use % not per cent

5. In results part, I would use column not Column. 

6 .Separate Discussion and Conclusions into two parts. In Discussion you discuss each hypothesis and present the result in comparison with other studies. In Conclusions, present theoretical and practical implications of your research, the novelty, the limitations and future research directions

7. Apendix A and B look like poor quality images. I think they should be presented as text or table to be clearer. Because now the text looks of poor quality being an enlarged image I think

I really liked the fact that you put them there. 

8. In the Discussion, you should add a few more references from 2021-2022.

In conclusion (mine, not yours), I really appreciate the professionalism with which the paper was written. 

Wish you all success in your careers. 

Author Response

The authors really appreciate all comments from the reviewer. All suggestions are very helpful in enhancing overall quality of the paper. Thus, authors have thoroughly amended the paper according to each, and every, comment/suggestion. Details of each point of revisions are described in the attachment.

Reviewer 2 Report

The paper is well-written and novel in its matter. Please find here some suggestions for improving it.

- Concept of CS I'm not convinced is embedding CSR, rather it is the contrary. Please consider that over both concepts there is sustainable development.

- From the abstract and introduction state which is the sample and the context in which you have collected the data (Thailand?). Please consider also in the literature review and in data analysis that this issue is having a strong influence on CSR provisions and practices

- if the focus is students, this must be mentioned from the beginning. Also the age is highly influencing data on consumers behaviour

- Please present the statistical model (the formula and not just the graphical representation) and dedicate some sentences on its description. Hypotesis are not sufficient, the model representation is nedded. 

- more space in the literature review should be dedicated to the signaling theory, because you are applying (correctly) that theory in the data discussion.

Author Response

(The authors gave the same response as above.)
